# Chemical Composition in Juvenile and Mature Wood of Branch and Main Trunk of *Leucaena leucocephala* (Lam.) de Wit

**DOI:** 10.3390/plants12233977

**Published:** 2023-11-26

**Authors:** Pramod Sivan, Karumanchi S. Rao, Kishore S. Rajput

**Affiliations:** 1Department of Chemistry, Division of Glycoscience, KTH Royal Institute of Technology, Albanova University Center, SE-10691 Stockholm, Sweden; 2Department of Biosciences, Sardar Patel University, Vallabh Vidyanagar 388120, Gujarat, India; kayesrao@yahoo.com; 3Department of Botany, The Maharaja Sayajirao University of Baroda, Vadodara 390002, Gujarat, India; ks.rajput15@yahoo.com

**Keywords:** *Leucaena*, wood, chemical composition, cell wall polysaccharides, lignin

## Abstract

Secondary growth is the most dynamic developmental aspect during the terrestrialization of plants. The development of secondary xylem tissue composed of thick-walled cells with characteristic changes in its structure and chemistry facilitates the growth and development of woody plants. In the present study, the chemical composition of the secondary xylem of juvenile and mature wood from the branch and main trunk of *Leucaena leucocephala*, has been investigated and the differences established. The biochemical analysis of different cell wall components in the mature wood of the main trunk revealed high holocellulose and α-cellulose and less lignin content in the juvenile wood while its syringyl/guaiacyl (S/G) ratio was less than for the mature wood. As compared to the branch xylem, concentration of cell wall polysaccharides and lignin content was higher in both juvenile and mature wood collected from the main trunk. Thioacidolysis and GC-MS analysis of wood lignin from juvenile and mature wood showed that an increased concentration in lignin content in mature wood is associated with a corresponding increase in S/G ratio. The structural information of the acetylated lignin was investigated by ^1^H NMR spectroscopy. Our results indicate that the mature wood from the main trunk is superior in pulp yielding and lignin degradability as compared to the juvenile wood of the branch and trunk.

## 1. Introduction

Developmental changes in the cell wall during secondary growth is one of the major events determining the growth and development of woody plants. The importance of a secondary vascular system in trees is mainly due to its thick-walled lignified cell wall components. Lignin is the second-largest biomass produced by land plants, acts as a long-term sink for atmospheric carbon and is increasingly exploited as an environmentally cost-effective renewable source of energy [1]. The complex chemical composition of wood (cellulose, hemicellulose, lignin, and pectin) makes it an ideal raw material for various purposes such as timber, paper, and pulp etc. However, the specific application of wood is determined by its physical and mechanical properties, which in turn depend upon its chemical composition [2]. Consequently, in the past years, the demand of wood from tree species as a source of paper and pulp has increased significantly because of the ability of trees to produce large amounts of biomass in a short time—though this is very dependent on location, climate, species, growth season etc. Recently, this trend has increased through genetic modification of trees to obtain wood with desirable chemical composition [3].

It is well known that wood shows considerable variation within the tree in its structural and mechanical properties, from center to outward and from the top to the bottom and juvenile and mature phases of the life cycle [4,5]. Variation patterns within the tree are closely associated with the characteristics and location of the juvenile wood and its relative proportion to mature wood [6]. Specific differences between wood formed during juvenile and mature growth phases are described for many species, but most studies focus on anatomical structure and mechanical properties [6,7]. Since chemical composition is directly related to the physical and mechanical properties of wood, recent studies have also been conducted on the chemical properties between juvenile and mature wood. The lower durability of the juvenile wood compared to mature wood has been correlated with a high concentration of extractives in *Tectona grandis* [8] and a lower content of phenolics and flavonoids in *Robinia pseudoacasia* (black locust) [9]. However, some authors found no relation between extractives and durability in black locust [10,11]. All these reports indicate that there is a positive correlation between chemical composition and utilization of wood for specific purposes. Therefore, it is necessary to know the suitable growth phase of the wood for its specific commercial utility and timber harvesting.

In both temperate and tropical regions, *Eucalyptus* wood has been the major hardwood tree species as a source of pulp and paper. However, in recent times, *Leucaena leucocephala* has emerged as an alternative hardwood species in the tropics and subtropical regions. It has tremendous value as a raw material for paper industry due to its high holocellulose and low silica content [12]. It is reported to grow in poor soils [13] where it has been found to provide an effective cover to leaching of heavy metals from mine tailings to the neighboring environment. Besides its fast-growing nature and great advantages of providing remediation for environmental problems, its suitable chemical composition for paper and pulp; *Leucaena* has become a popular multi-purpose tree in tropical and sub-tropical countries. Although there are reports on the difference in chemical composition between different varieties of *Leucaena leucocephala* [14,15], variations in the chemical composition within the tree have not yet been studied. Moreover, the *Leucaena* trees have been harvested at an interval of three years after planting and in addition to the main trunk, the branches (0.5–5 cm thickness) could also be used as a paper-making raw material [15]. Hence it is equally important to know the wood’s chemical composition in the branches and main trunk to determine better a harvesting period for its utilization. Keeping this in mind, in the present study, we have investigated the chemical composition in the juvenile and mature wood from the branches and main trunk of *Leucaena leucocephala.* Our study indicates wood maturation occurs through characteristic changes in the chemical composition of the cell wall resulting in variation in the quantity and composition of holocellulose, hemicelluloses and lignin.

## 2. Results

### 2.1. FTIR Analysis of Wood Powder (WP) and Klason Lignin (KL)

The extract-free, milled wood powder and Klason lignin residue extracted from the same wood powder sample were subjected to infrared spectroscopy analysis to detect the fingerprint regions corresponding to their chemical composition. FTIR analysis of wood powder (WP) of juvenile and mature wood from branch and trunk wood showed prominent peaks in the fingerprint region of 1800–600 cm^−1^ (Figure 1). The peaks were numbered and assigned to chemical compounds (Appendix A) according to the published literature [16,17]. Most of the peaks in the WP represent major cell wall polysaccharides such as cellulose (peaks 1, 2, 9, 12, and 14), hemicelluloses (Peak 3, 8, 13, 14) and pectins (Peak 17). When the Klason lignin was subjected to FTIR analysis, the absorbance in the polysaccharide region from 1200–900 cm^−1^ was strongly diminished compared to that of WP (Figure 1). The peaks 4, 5, 6, 7, 10, 11 and 15 (1627–1629 cm^−1^, 1541–1542 cm^−1^, 1505–1506 cm^−1^, 1459–1461 cm^−1^, 1322–1323 cm^−1^, 1284–1285 cm^−1^ and 1010 cm^−1^) corresponding to S and G lignin types became more prominent in Klason lignin. Peak 5 at 1599–1601 cm^−1^ in the WP spectra arose due to aromatic skeleton vibration plus C=O stretch in the SG-type lignin was shifted to position 1541 in Klason lignin spectra. A shift in position of the peak at 1629 cm^−1^ attributed to many non-cellulosic polysaccharides was also noticed in the KL spectra.

### 2.2. Holocellulose and α-Cellulose Content

Variation was noticed in holocellulose content in the juvenile wood between the branch and main trunk xylem (Table 1). The latter showed more holocellulose than the branch. After wood maturity, the trunk wood xylem showed a decline in holocellulose while branch wood did not show significant variation between juvenile and mature wood (Table 1). Variation was also noticed in holocellulose content in the juvenile wood between branch and main trunk xylem. The highest α-cellulose content was found in the juvenile wood from the main trunk. After maturity, both branch and main trunk xylem showed a decrease in α-cellulose content. However, the main trunk contained more α-cellulose than the branch wood during both the developmental stages.

### 2.3. Sugar Composition Analysis

The acid methanolysis data revealed that sugar units in hemicelluloses and pectin were higher in mature wood compared to those of juvenile wood of *Leucaena* (Table 2). The differences in sugar units were more apparent between branch wood and trunk wood while relatively fewer differences were noticed between mature and juvenile wood of the same origin. The major neutral sugar units such as xylose and galactose as well as acidic sugar units such as glucuronic acid and galacturonic acid associated with hemicelluloses and pectins increased substantially after wood maturation. On the other hand, 4-O-methyl glucuronic acid was relatively lower in mature wood compared to that of juvenile wood.

### 2.4. Lignin Content

Juvenile and mature wood samples from main trunk and from branch wood were subjected to Klason lignin estimation (Table 3). Lignin content was higher in the mature wood of branch (23.33%) and main trunk (24.03%) compared to that of juvenile wood (20.43% and 21.3%, respectively, in branch and main trunk). Main trunk xylem also contained slightly more lignin (Table 3).

### 2.5. Lignin Monomeric Composition

The results of the gas chromatographic separation of thioacidoltytic monomers from juvenile and mature wood lignins are displayed in Figure 2. The S/G ratios in the juvenile and mature wood from the branch and main trunk of *Leucaena* evaluated using the sum of the G- and S-types of thioethylated monomers are listed in Table 4. In both branch and main trunk, the S/G ratio was increased in mature wood as compared to juvenile wood. No significant variation was found in the S/G ratio between juvenile wood from branch and main trunk.

### 2.6. Structural Analysis of Lignin

The structural characteristics of acetylated Bjorkman lignin of *L. leucocephala* were analyzed with 1H NMR spectroscopy. The basic interpretation of spectra was carried out based on previous data available from the model compounds and hardwood lignin. The results are presented in Figure 3 and Table 5. The integrals of all signals between δ6.8 and δ7.02 belonged to aromatic protons and the signals from 3.4 to 4.23 were attributed to aliphatic protons. The strong signals at the aromatic region δ7.0 to δ7.2 and δ6.84 to δ6.93 were assigned to aromatic protons in guaiacyl (G) and syringyl (S) lignin respectively [18]. In bagasse soda lignin, 1H NMR spectra of acetylated lignin showed syringyl protons signal occur between 6.28 and δ6.80 while the guaiacyl proton signal occurs between δ6.80 and δ8.00 [19]. The arylglycerol-β-aryl ether inter-monomeric linkages can be erythro (E) or threo (T) forms and the E/T ratio is an important structural feature of lignin [20]. The signals at δ6.93 and δ6.84 might be produced due to Hα in the erythro and threo forms of aryl-glycerol ether inter-monomeric linkages. Strong signals at δ2.0–δ5.1 originating from methoxy protons and β-O-4 structure were also observed in the spectra. Among erythro and threo forms of these structures, only a single strong signal was noticed in the erythro form (δ4.29), whereas several signals (4.65, 5.03, 3.81, 3.71, and δ2.10) correspond to the threo form of β-O-4 structures observed in the 1H NMR spectra, indicating that *Leucaena* lignin may be dominated by threo forms of β-O-4 structures. The mature wood showed relatively more methoxy protons as compared to juvenile wood (Table 6).

### 2.7. Composition of Carbon, Hydrogen and Nitrogen Elements in Lignin

The elemental analysis revealed that the lignin from mature wood is characterized by relatively higher hydrogen and nitrogen content compared to those of juvenile wood (Table 7).

## 3. Discussion

### 3.1. FTIR Analysis of Wood Powder (WP) and Klason Lignin (KL)

The chemical fingerprinting of composition in the wood cell wall and its variation in Klason lignin was analyzed by FTIR spectroscopy. Variation was noticed in the form of absence of peaks, intensity variation or as a shift in peak positions. Many of the prominent peaks in the FTIR spectra of WP were specific to cellulosic polysaccharides i.e., peaks 1, 2, 9, 12 and 14 (at 3410 cm^−1^, 2906 cm^−1^, 1375 cm^−1^, 1161 cm^−1^ and 1056 cm^−1^) were diminished in the spectra of the Klason lignin of the same samples. This shows that the cellulosic polysaccharides have been removed from the wood powder during acid hydrolysis. Peak 13 (at 1111 cm^−1^) in the WP belonging to xyloglucan was also removed in the spectra of Klason lignin. On the other hand, peaks 3 and 14 (at 1721–1738 and 1056–1069 cm^−1^) corresponding to hemicelluloses were observed in the spectra of both WP and KL. This may be due to their cross-linkage with lignin which results in them not being removed by acid treatment. It is possible that the xylan associated with lignin might have resulted in a peak at 1264 cm^−1^ while this substituent may be removed in Klason lignin leading to a peak at 1284 cm^−1^ corresponding to guaiacyl lignin. On the other hand, the peak numbers 6, 7 and 10 (1505–1510, 1459–1461 and 1322–1329 cm^−1^) belonging to G- and GS-type lignin did not show any peak shift in the FTIR spectra of WP and KL—indicating that they may represent pure lignin peaks without interference from any substituent on its aromatic ring. The peaks 3 and 11 in the FTIR spectra of WP at 1738 and 1274 cm^−1^ corresponding to unconjugated carbonyl and carboxyl groups in xylan [24,25] were shifted to position 1716 and 1284 cm^−1^, respectively, which were formed due to conjugated acids and un-conjugating carbon groups in lignin [26] and guaiacyl ring broadening in G lignin [27] respectively. We hypothesize that this shift might be due to the cross-linkage between xylan and G-lignin. The shift in wavenumber can be due to the inductive effect of substituents in the aromatic ring system of lignin [26]. The peak at 1629 cm^−1^ is very complex, being attributed to many non-cellulosic polysaccharides such as pectin, xylan, galactan, arabinogalactan etc., [28] and the aromatic skeleton vibration in the highly conjugated C=C bond of lignin [29]. In the present study, this peak also showed a shift in its position in the spectra of WP and KL indicating the complex lignin–hemicellulose interaction.

### 3.2. Content and Composition of Cell Wall Polysaccharides

A decrease in holocellulose content was apparent during the maturation of main trunk wood. A higher α-cellulose content in the main trunk wood compared to branch wood suggests variation exists during wood maturation between tree organs. The holocellulose and α-cellulose content in the three-year-old *Leucaena leucocephala* from India was reported to be 76.4 and 44.6, respectively [15]. These values are similar to those found in the juvenile wood from the main trunk. Nazri et al. [29] have reported high holocellulose content in the juvenile wood from the trunk and lowest value in the 16-year-old trunk of *Leucaena*. Therefore, the decreased content in holocellulose and α-cellulose may be a part of mature wood formation in both branch wood and the main trunk. The increase in cell wall thickness has been a conspicuous feature during the transition from juvenile to mature wood in *L. leucocephala* [3]. Therefore, probably, the increases in sugar units in hemicellulose and pectins may be associated with increase in the volume of secondary wall thickness in the mature wood.

### 3.3. Lignin Content and Monomer Composition

Our study also shows an increase in lignin content during wood maturation in *Leucaena*. Dunisch et al. [9] reported an increase in lignin in the mature wood compared to juvenile wood of *Robinia pseudoacasia*. These authors pointed out that the variation in lignin content during the juvenile to mature wood growth phase may be associated with its cell wall ultrastructure. The ultrastructure of cell walls formed during juvenile growth phase differs significantly from that of cell walls formed by an older cambium [30,31,32]. The microfibrillar orientation in the S2 layer of the juvenile cells (higher microfibrillar angle) is related to lower lignin content of the juvenile wood [11].

The effect of plant maturity on lignin and forage digestibility has been reported in grass species [33,34,35]. However, little information is available on lignin monomeric composition in hardwood species during juvenile and mature wood growth phases. Chen et al. [35] studied the changes in lignin and its monomeric composition in grass during different developmental stages and found that syringyl lignin units, and the ratio of syringyl: guaiacyl units in lignin, increases with progressive maturity of the stem. The present study also shows an increase in the S/G ratio in mature wood compared to juvenile wood in the branches and main trunk of *Leucaena*. The content and chemical structure of wood components (particularly lignin content) and its composition in terms of its guaiacyl (G) and syringyl (S) moieties are important parameters in the paper and pulp industry in terms of delignification rates, chemical consumption, and pulp yield [36]. The higher reactivity of S lignin compared to G lignin in alkaline systems is known [37] and therefore the lignin with low S/G ratio in hardwoods affects pulping efficiency. This is mainly because of the highly branched structure of guaiacyl lignin due to rapid polymerization and they form intermolecular coupling mainly through condensed bonds. On the contrary, the syringyl lignin is formed by extensive coupling of less-condensed β-O-4 linkages. Since mature wood in *Leucaena* shows a high S/G ratio which is positively correlated with lignin degradability in pulping process and high α-cellulose content, it would be judicious to collect wood after its maturity. The anatomical changes during the transition from juvenile to mature wood formation in *Leucaena* showed that the wood attains its mature wood characteristics within 5–6 years of growth which is comparable with those of 15-year-old trunk wood xylem [3]. Therefore, the data on chemical composition in the present study indicate that the harvesting of wood after six years of wood growth may provide better lignin degradability and pulp yield.

### 3.4. Structural Analysis of Lignin

The 1H NMR spectra also provide information on the occurrence and distribution of different types of structural elements of lignin such as β-5, β-β, non-cyclic benzyl ether, phenolic group etc., [18]. The β-β structures in hardwood lignin are primarily of syringaresinol type and the acetate of this compound generally shows a signal near δ5.7 (Hα) in the spectra [38,39]. Lundquist [18] suggested that the signal formed at δ5.76 in the acetylated birch lignin could be produced due to the presence of minor number of structural elements corresponding to dimer formation. In *Leucaena* lignin, presumably, the signal found in the δ5.72 might be associated with structures related to dimer formation or due to other types of lignin units carrying benzyl alcoholic groups [18].

Phenolic acetate groups in guaiacyl, syringyl as well as p-hydroxyphenyl units contribute to the lignin peak at δ2.3 [18]. *Leucaena* lignin showed a signal at 2.31 in the spectra corresponding to phenolic acetate group structures (Figure 3). The phenolic group of the guaiacyl type is larger than the number of phenolic groups of the syringyl type [18,40]. Therefore, the strong signal strength of the phenolic acetate group suggests the abundance of guaiacyl lignin and this further supports our thioacidolysis data of a low S/G ratio in *Leucaena* wood lignin. Previous studies on 1H NMR spectra of Bjorkman lignin have revealed the occurrence of few percent aryl glycerol β-aryl ether structures carrying ether groups at the α-position resulting in a signal at δ5.4 [18,38,41]. *Leucaena* lignin also showed a signal at δ5.39 indicating the presence of aryl-ether groups raised due to non-cyclic benzyl aryl ether during hydrolysis of wood lignin [42].

The mature wood in *Leucaena* was also characterized by a relatively higher level of methoxy protons as compared to juvenile wood (Table 6). It is well known that the monolignol composition in hardwood lignin is distinguished by a high S/G ratio and syringyl lignin is characterized by a greater number of methoxy units compared to that of guaiacyl lignin monomers. Akiyama et al. [43] suggested that the decrease in methoxy group content in the poplar tension wood lignin is due to a corresponding decrease in syringyl/guaiacyl ratio. The high S/G ratio in mature wood revealed by thioacidolysis data also suggests the incorporation of more syringyl units during wood maturation and therefore we assume that the relatively higher proportion of methoxy protons in the mature wood may be associated with the increase in syringyl lignin units in the cell wall. The elemental analysis shows relatively higher hydrogen and nitrogen content in the mature wood lignin compared to those of juvenile wood, suggesting the change in the molecular structure of lignin associated with its monomeric composition during wood maturation in *L. leucocephala.*

### 3.5. Variation in Chemical Composition between Branch Wood and Main Trunk Wood

The chemical composition of wood has a major role in determining the growth and developmental aspects. Therefore, understanding the wood chemistry of different plant organs with specific function is important to unravel the functional dynamics of wood. Our study indicates that the chemical composition in two different organs (main trunk and branch) shows a similar pattern of variation in *L. leucocephala.* However, we also noticed some variation in chemical composition between the same developmental stages of two organs. The branch wood is characterized by relatively low cellulose content, low lignin content with high S/G ratio compared to main trunk wood, suggesting that the chemical variation might be related to the functional attributes of these organs. This variation could be due to relatively higher sapwood in the branch wood compared to that of the main trunk [44]. The special growth patterns such as plagiotropic growth in branches in response to gravity can also influence the structure and chemistry of wood [45]. Irrespective of this variation related to the functional dynamics of two different wood traits within the trees, the holocellulose content in the range of 68–70% in the branch wood suggests its potential for valorization into biorefinery applications [46,47]

In summary, the chemical composition of juvenile and mature wood of *Leucaena* showed specific differences between branch and main trunk. Both the juvenile and mature wood in main trunk xylem showed a higher holocellulose and α-cellulose content, less lignin and low S/G ratio compared to that of branch wood. The S/G ratio and Klason lignin are positively correlated as the mature wood shows high lignin content and S/G ratio. Both the juvenile and mature wood in main trunk xylem showed a higher quantity of holocellulose, α-cellulose and S/G ratio than that of branch wood, indicating a higher pulp value for trunk wood xylem than branches. The acid methanolysis data revealed that the sugar units in hemicelluloses and pectin were higher in mature wood compared to those of juvenile wood of *Leucaena* while relatively fewer differences were noticed between mature and juvenile wood of the same origin. The 1H NMR spectra indicated that the *Leucaena* lignin may dominated by β-O-4 structures and mature wood with relatively more methoxy protons compared to that of juvenile wood. The information obtained could be useful for the genetic modification of the *Leucaena* and also to determine a better harvesting age of the trees for better pulp yield.

## 4. Materials and Methods

### 4.1. Plant Material

Six-year-old trees of *Leucaena leucocephala* (Lam.) de Wit were harvested in February 2009 from Sardar Patel University, Botanical Garden, Gujarat, India. Two wood discs having a girth of 35 cm were collected 2 m above ground-level from the vertically growing trees to avoid the reaction of wood xylem. Similar wood discs were also collected from the main branches growing vertically and having similar girth (about 35 cm) from 15-year-old trees growing in the same location. The juvenile wood was collected from the inner core of the wood disc near the pith, while the mature wood samples were collected from the periphery of the disc. Wood pieces were frozen in liquid nitrogen for 1 h and subsequently dried for 72 h at −80 °C. The wood powder (WP) was obtained by pulverizing the samples in a Wiley mill to pass through an 80 µm sieve. The powdered wood (WP) was then sequentially extracted with distilled water, ethyl alcohol, alcohol:toluene (1:1) and acetone (Merck, Germany) in a Soxhlet extractor (Shiva Scientific Glass Ltd, New Delhi, India)and dried to obtain extractive free xylem residue (EXR).

### 4.2. FTIR Spectroscopy

Wood powder (WP) and Klason lignin (KL) were used for FTIR analysis. KBr pellets for IR spectroscopy were prepared using macrotechnique (13 mm ø pellet; Ca. 1.5 mg sample with 350 mg KBr). The spectra were recorded with the FTIR spectrometer with a TGS detector (Perkin Elmer, Spectrum GX, Shelton, USA) at a resolution of 4 cm^−1^ for 32 scans in the range from 600 to 4000 cm^−1^; background spectra of a clear window were recorded prior to the acquisition of sample spectra. The spectrum of the background was subtracted from spectra of the sample before conversion into absorbance units. For each sample, three different sub-samples were analyzed and averaged to give a mean spectrum per individual sample.

### 4.3. Determination of Holocellulose and α-Cellulose

The holocellulose and α-cellulose were determined using a micro-analytical method developed by Yokoyama et al. [48], in which 200 mg of EXR was weighed into a 15 mL round bottom flask and placed in water bath at 90 °C. The reaction was initiated by the addition of 1 mL of reaction mixture (400 mg of 80% sodium chlorite + 4 mL distilled water + 0.4 mL acetic acid). An additional 1 mL reaction mixture was added every 30 min, and the samples were removed to a cold-water bath after 2 h. The sample was then filtered through a course sintered glass filter (Whatman GD 1UM), washed with deionized water (3 × 50 mL), dried in an oven at 105 °C and holocellulose content was determined gravimetrically. For determination of α-cellulose, 50 mg of dried holocellulose was weighed into a reaction flask and allowed to equilibrate for 30 min, 4 mL of 17.5% sodium hydroxide was added and allowed to react for 30 min and then 4 mL of distilled water was added. The sample was macerated for 1 min, allowed to react for 29 min and then filtered through a sintered glass filter. Following a 5-min soaking in 1.0 M acetic acid, the sample was washed with 90 mL of distilled water (3 × 30 mL) and dried overnight. The α-cellulose was determined gravimetrically.

### 4.4. Determination of Lignin Content

Lignin analysis was carried out on dry extract-free wood powder and ground to pass through a 180 micron sieve before exhaustive solvent extraction (2:L1 (*v*/*v*) toluene: ethanol, ethanol and water). The lignin content in the juvenile and mature wood was determined by the Klason lignin method [49]. Acid-soluble lignin content was determined by measuring the UV absorption at 205 nm using an extinction coefficient of 110 g/1 cm^−1^ of H_2_ SO_4_ hydroxylate.

### 4.5. Thioacidolysis

Thioacidolysis was carried out according to Lapierre et al. [50]. The reagent was prepared by introducing 2.5 mL of BF3 etherate (Aldrich) and 10 mL of ethanethiol EtSh (Aldrich) into a 100 mL flask and adjusting the final volume to 100 mL with dioxane. A mixture of the sample (12 mg) and 12 mL of reagent were placed in tube fitted with a Teflon-lined screw cap. Thioacidolysis was performed at 100 °C (oil bath) for 4 h with an occasional shaking. The cooled reaction mixture and the washings with water were combined and the mixture was poured over 1 mL dichloromethane (Fluka, Germany) including internal standard (0.50 mg tetracosane from Sigma, Germany). After adjusting the pH of the aqueous phase to pH 3–4 with 0.4 M sodium carbonate aqueous solution, the aqueous phase was then extracted with dichloromethane (20 mL × 3). The combined organic extracts were dried over Na_2_SO_4_ and the solvent was evaporated under reduced pressure at 40 °C in a rotary evaporator. The residue was re-dissolved in dichloromethane (1 mL). The thioacidolysis products (7 µL) were silylated with 50 µL of N, O-bis (trimethylsilyl) trifluroacetamide (BSTFA, Sigma, Germany) and 5 µL of pyridine (Sigma, Germany) in a 200 µL GC vial with a Teflon-lined screw cap and kept at room temperature overnight. The silylated products were separated by gas chromatography using a silicon-based capillary column (30 m × 0.25 µm × 250 µm) and each peak was identified by GC-MS. The temperature program of the GC was increased at a rate of 5 °C/min from 100 °C/280 °C and then the final temperature was maintained for 60 min.

### 4.6. Determination of Hemicelluloses and Pectin

The sugar units in hemicelluloses and pectins were determined by gas chromatography (GC) analysis after acid methanolysis [51]. A 10 mg of EXR was weighed in a pressure-resistant pear-shaped flask with a screw cap. Then, 2 mL of a 2 M solution of HCl in anhydrous methanol was added to the samples and they were kept in an oven at 100 °C for 5 h. After cooling to room temperature, 150–200 µL of pyridine was added and then 4 mL of methanol containing sorbitol (0.1 mg/mL) was added as internal standard. The sample was mixed and 1 mL of the clear solution was transferred into a test tube. The samples were then evaporated under a stream of nitrogen at 50 °C and further dried in a vacuum oven at 40 °C until nearly dry. The samples were then silylated by adding 150 µL of pyridine, 150 µL hexamethyldisilazane (HMDS) and 70 µL trimethylchlorosilane (TMCS). The samples were vigorously shaken and left to stand overnight. A part of the clear solution was transferred to GC auto-sampler vials. Then, 1 µL of silylated samples were injected via split injector (260 °C, split ratio 1:20) into a 25 m × 0.25 mm × 250 µm thick HP-1 column. The column temperature was 100 °C to 175 °C (4 °C/min) followed by 175 °C to 290 °C (12 °C/min). A calibration solution containing equal amounts of analytical grade arabinose, xylose, galactose, glucose, mannose, rhamnose, glucuronic acid and galacturonic acid (0.1 mg/mL) was subjected to acid methanolysis and analysis at the same conditions as the samples to obtain correction factors for losses of sugar residue during methanolysis. Analysis was performed in triplicate for each sample. Total amount of hemicelluloses and pectins were calculated from the sugar units considering the removal of water (according to the respective coefficients of pentose and hexose sugars).

### 4.7. Isolation of Milled Wood Lignin (Bjorkman’s Lignin)

The extraction and purification of milled wood lignin from the extractive free wood powder was carried out according to the Bjorkman’s method [52]. The juvenile and mature wood collected from the main trunk and branch wood disks were subjected to milled wood powder preparation. Lignin was extracted from 10 gm of extractive free wood powder (for each sample) by suspension in dioxane–water (9:1, *v*/*v*) and stirring for three weeks. The wood meal was then separated and discarded from dioxane solution by centrifugation. The dioxane solution was evaporated under reduced pressure. The Bjorkman lignin was prepared by purifying the residue with 80% acetic acid followed by a mixture of 1, 2-dichloroethane and ethanol (2:1, *v*/*v*). This solution was washed with diethyl ether 2–3 times and once with hexane. The purified lignin was then dried in air, then in an oven under vacuum.

### 4.8. Acetylation of Bjorkman Lignin

The acetylation was performed according to Gosselink et al. [53]. The Bjorkman lignin (100 mg) was acetylated by reaction with 2 mL of pyridine:acetic acid mixture (1:1 *v*/*v*) at room temperature overnight using a magnetic stirrer. Then, 2.5 mL of ice-cooled 100% methanol was added, the product was evaporated at reduced pressure to dryness and suspended in toluene and again evaporated to dryness (three times). The final residue was dissolved in methanol and evaporated to dryness.

### 4.9. ^1^H NMR Analysis

Acetylated Bjorkman lignin (10 mg) was dissolved in 0.5 mL of CDCl3. The 1H NMR spectra were recorded with an Avance III 400 MHz FT-NMR spectrometer (Brukar BioSpin GmbH, Rheinstetten, Germany). Proton signals were integrated from base line and referred to the integral signal of the methoxyl proton for the quantification of aliphatic and phenolic protons. The ratio of aromatic to methoxyl protons was determined by integrating signals at 6.8 to 7.2 and 3.4 to 4.3, respectively [19].

### 4.10. Statistical Analysis

ANOVA test (Tukey method) was carried out to determine statistically significant differences of anatomical parameters at a 0.05 confidence level using Sigmastat software (Version 3.5, San Jose, California, USA).

### 4.11. Elemental Analysis

Carbon, hydrogen and nitrogen contents of Bjorkman lignin were determined using a Thermo Finnigan Flash EA 1112 CHN analyzer (Thermo Finnigan Italia, Rodano, Italy).

## Figures and Tables

**Figure 1 plants-12-03977-f001:**
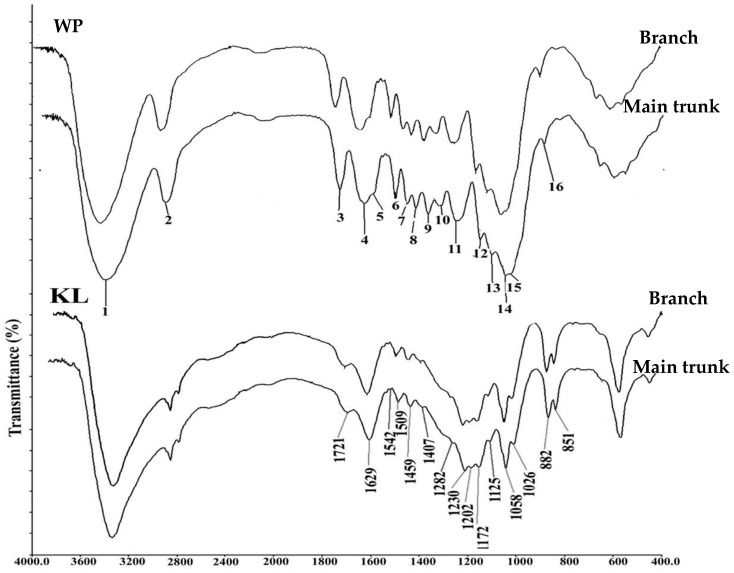
FTIR spectra of wood powder (WP) and Klason lignin (KL) from juvenile wood of *Leucaena leucocephala*.

**Figure 2 plants-12-03977-f002:**
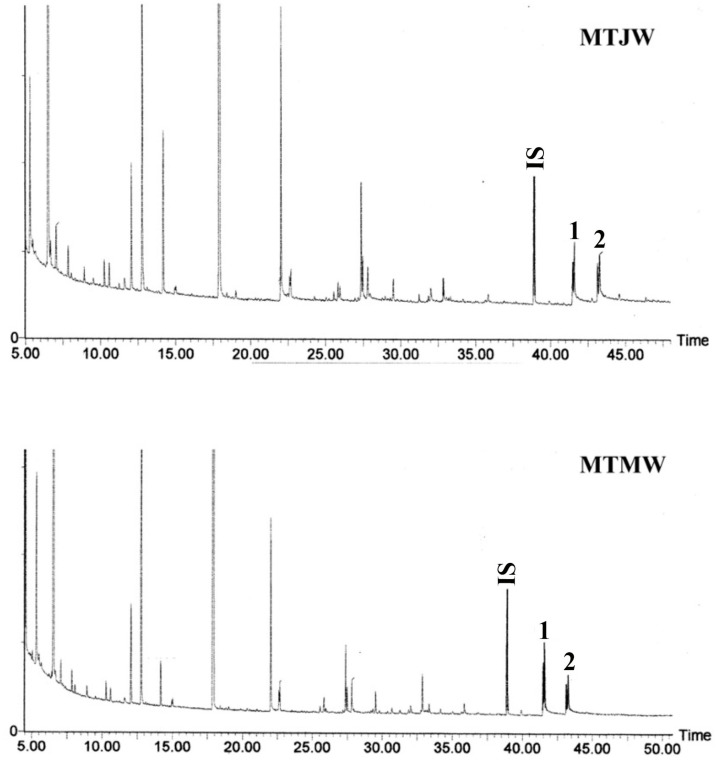
Gas chromatographs of TMS thioacidolysis products recovered from juvenile (MTJW) and mature wood (MTMW) from the main trunk of *Leucaena leucocephala.* Numbered peaks correspond to TMS thioacidolysis products (C_6_C_3_ arylglycerol-β-aryl ether) 1. G-CHSEt-CHSEt-CH2SEt erythro/threo, 2. S-CHSEt-CHSEt-CH2SEt erythro/threo. G: Guaiacyl, S: Syringyl, IS: Internal standard (Tetracosane).

**Figure 3 plants-12-03977-f003:**
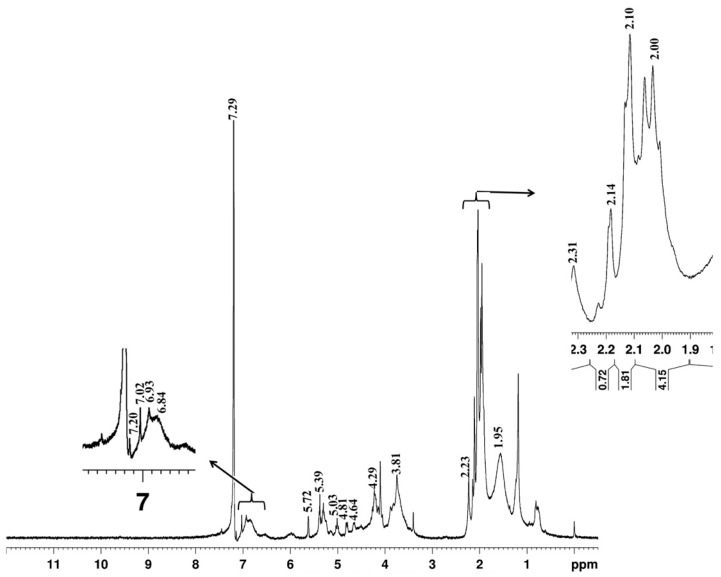
^1^H NMR spectrum of acetylated milled wood lignin from *Leucaena leucocephala.* Arrows indicates the enlarged regions corresponds to aromatic protons in G-lignin, S-lignin, aliphatic acetate and CH3O (threo) β-O-4 structures.

**Table 1 plants-12-03977-t001:** The holocellulose and α-cellulose content in the juvenile and mature wood from branch and main trunk of *Leucaena leucocephala*.

Sample	Juvenile Wood		Mature Wood	
	Branch	Main Trunk	Branch	Main Trunk
Holocellulose	68.50 ± 1.00	* 74.50 ± 1.50	70.00 ± 1.20	* 67.50 ±2.00
α-cellulose	48.00 ± 1.00	* 52.00 ± 1.20	46.00 ± 1.00	* 50.00 ± 2.00

Mean values from the two trees × two samples per each tree are given in the table. * Significant difference between juvenile and mature wood following ANOVA test (α = 0.05).

**Table 2 plants-12-03977-t002:** Amount and composition of sugar units in hemicelluloses and pectins, expressed as mg/g dry wood ± two times the standard deviation for juvenile and mature wood from branch (BJW and BMW) and main trunk (MTJW and MTMW) of *Leucaena leucocephala*.

Component	BJW	MTJW	BMW	MTMW
Arabinose	3.13 ± 0.40	3.22 ± 0.24	* 3.60 ± 0.50	* 3.80 ± 0.54
Xylose	101.10 ± 2.8	106.20 ± 3.46	111.20 ± 5.00	112.30 ± 5.40
Galactose	9.20 ± 1.00	11.50 ± 1.02	* 14.40 ± 1.8	* 13.90 ± 2.00
Glucose	42.00 ± 16.10	40.60 ± 20.40	43.90 ± 18.80	42.80 ± 21.20
Mannose	3.10 ± 0.10	3.35 ± 0.25	* 3.85 ± 0.18	* 3.90 ± 0.28
Rhamnose	5.10 ± 0.32	4.21 ± 0.26	* 2.30 ± 0.40	* 2.21 ± 0.14
GlCA	0.39 ± 0.20	0.53 ± 0.36	* 0.61 ± 0.21	* 0.68 ± 0.24
4-O-MeGlcA	13.10 ± 2.0	13.30 ± 1.90	* 9.50 ± 2.20	* 9.40 ± 1.60
GalA	13.60 ± 3.20	12.00 ± 3.50	15.20 ± 2.90	* 14.90 ± 2.40
Total	190.82 ± 20	194.91 ± 24.9	* 204.46 ± 23.2	* 203.79 ± 18.4

* Significant difference between juvenile and mature wood following ANOVA test (α = 0.05). GlcA = Glucuronic acid; GalA = Galacturonic acid; 4-O-MeGlcA = 4-O-methyl glucuronic acid.

**Table 3 plants-12-03977-t003:** Klason lignin content (%) in juvenile and mature wood from branch and main trunk of *Leucaena leucocephala*.

Sample	Juvenile Wood	Mature Wood
Branch	20.43 ± 0.50	* 23.33 ± 0.40
Main trunk	21.29 ± 0.60	* 24.03 ± 0.50

Mean values from the two trees × two samples per each tree are given in the table. * Significant difference between juvenile and mature wood following ANOVA test (α = 0.05).

**Table 4 plants-12-03977-t004:** Monomeric composition of lignin from juvenile and mature wood from branch (BJW, BMW) and main trunk (MTJW, MTMW) of *Leucaena leucocephala*.

Sample	Monomeric Composition			
	Guaiacyl (G)	Syringyl (S)	S + G	S/G Ratio
BJW	490.62	400.92	900.50	0.82
MTJW	540.87	450.83	1000.60	0.83
BMW	760.53	670.28	1430.80	* 0.89
MTMW	800.85	720.12	1520.90	* 0.90

G, S, S + G: Yields of the thioethylated guaiacyl (G), Syringyl (S) and total (S + G) expressed as µmoles per gram of lignin. Mean values from the two trees × two samples per each tree are given in the table. * Significant difference between juvenile and mature wood following ANOVA test (α = 0.05).

**Table 5 plants-12-03977-t005:** 1H chemical shifts and signal assignments for acetylated lignin from juvenile and mature wood of *Leucaena leucocephala*.

Shift δ (ppm)	Assignments
7.29	Solvent (CDCl3)
7.02, 7.20	Aromatic proton in G-lignin [21]
6.84, 6.93	Aromatic proton in S-lignin [21]
5.39	Hα in β-5 structure and non-cyclic benzyl aryl ether [21]
5.03	Hα (threo) in β-O-4 structures [22]
4.81	Hα in β- β structure (methylene proton in cinnamyl alcohol units) [22]
4.64	Hβ in β-O-4 structures [22]
4.29	Hγ1 (erythro) in β-O-4 structures [22]
3.81	OCH3 (threo) in β-O-4 structures [22]
2.23	Aromatic acetate [23]
2.14	Aliphatic acetate (including some aromatic acetate) [23]
2.00, 2.10	CH3CO (threo) in β-O-4 structures [22]
1.95	Aliphatic acetate [23]

**Table 6 plants-12-03977-t006:** Methoxy (OH-OCH3) and aromatic (OH-Ar) proton content in the lignin of juvenile and mature wood of *L. leucocephala* determined by NMR spectroscopy methods (%, *w*/*w*).

	BJW	MTJW	BMW	MTMW
OH^-OCH3^	5.00	5.02	5.68	5.71
OH-^Ar^	1.00	1.01	1.07	1.10
OH-^OCH3/^OH-^Ar^ ratio	0.19	0.19	0.20	0.20

**Table 7 plants-12-03977-t007:** Elemental composition of isolated lignin from juvenile and mature wood of *L. leucocephala*.

	Carbon (%)	Hydrogen (%)	Nitrogen (%)
BJW	49.18	4.58	0.39
MTJW	48.45	4.65	0.40
BMW	45.06	5.34	0.51
MTMW	46.29	5.56	0.53

## Data Availability

The data presented in this study are available on request from the corresponding author. Original data collected and processed to obtain the results presented in this study are not publicly available due to their lack of interest.

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
