# Peer review of "Chemical Composition in Juvenile and Mature Wood of Branch and Main Trunk of Leucaena leucocephala (Lam.) de Wit"

_plants, 2023, doi:10.3390/plants12233977_

Round 1
Reviewer 1 Report
Comments and Suggestions for Authors
The authors conducted a chemical compositional analysis of Leucaena leucocephala, one of the expecting candidates as advantageous biomass resource. They paid extra attention to the difference in features of wood between juvenile and mature, and between branch and main trunk. They found several differences, i.e. α-cellulose content had tended to be higher in main trunk rather than a branch; lignin content was higher in mature wood than juvenile, and S/G ratio showed the same tendency with the lignin content. The research is properly conducted, including the methods used. The data is presented clearly. Their findings presented here provide basic chemical features of wood of this species, and expected to contribute to future development of effective harvesting period and/or industrial utilization of this wood.
1 I think several sentences are missing from the Introduction, because the 2nd paragraph suddenly begins with “closely associated…” and references 4 and 5 do not appear.
2 I think the sample name “CWR: cell wall residue” is puzzling and inappropriate. I strongly suggest to rename, so it would not cause any misunderstandings.
3 Description of two spectra in each CWR and KL seems to be missing in Fig. 1. Are they represent same type of sample with different individual, or different type of sample not mentioned in the text? (e.g. juvenile and mature woods of the main trunk)
4 In some part, the term “mature” wood is used, and in the other part the “adult” wood is used. Was this done for a reason? If not, can’t it be standardized?
5 I think the Discussion Section should be revised focusing on the following two points:
--- The present version is without subsection. In other words, readers are forced to read more than two pages-long sentences without stopping. Please make the manuscript more easily read.
--- Especially in this case, since the Results and the Discussion are separated, I think discussion combining several results from different analytical methods should be included. I am afraid that the present version basically discussing each result separately. (e.g. Is there any trend that can be noticed when three major components of wood are discussed together?)
Additional minor comments:
6 There are many small mistakes in writings, so close proofreading is the must: e.g. typos (which appears especially in the reference list) and mis-writings, mistakes in capitalization and italicization rules, not using superscripts and subscripts where needed, numbering order of the cited references, etc. Please check the sentences in line389 and line 395. Are they expressing what the authors wanted to say correctly? (I am suspicious that some essential words are missing from these sentences.)
7 Regarding the G peak and S peak tagged in Figure 2. Couldn't the descriptions would be more specific, namely the exact compound names?
8 I personally think the result and the discussion presented in the current manuscript is not enough to mention threo β-O-4 structure may dominate in Leucaena lignin.
9 Where is the result of lignin content determined by acetyl bromide methods?
Comments on the Quality of English LanguageBasically, the quality is well enough.
Author Response
- I think several sentences are missing from the Introduction, because the 2ndparagraph suddenly begins with “closely associated…” and references 4 and 5 do not appear.
Response: Thank you very much for finding this important correction. During final formatting, some sentences and its part went missing. The missing sentence and part of the sentence have been added
- I think the sample name “CWR: cell wall residue” is puzzling and inappropriate. I strongly suggest to rename, so it would not cause any misunderstandings.
Response: CWR have been replaces with WP (wood powder) as it reflects the general nature of sample.
- Description of two spectra in each CWR and KL seems to be missing in Fig. 1. Are they represent same type of sample with different individual, or different type of sample not mentioned in the text? (e.g. juvenile and mature woods of the main trunk)
Response: A brief description of the WP and KL have been added in the results. The preparation details of them are already present in the material and methods part. The klasson lignin was prepared from the same wood samples. These details are now included in the results.
- In some part, the term “mature” wood is used, and in the other part the “adult” wood is used. Was this done for a reason? If not, can’t it be standardized?
Response: Adult wood and mature wood are same. As per the suggestion from reviewer, now the term ‘mature wood’ have been used throughout the manuscript.
- I think the Discussion Section should be revised focusing on the following two points:
--- The present version is without subsection. In other words, readers are forced to read more than two pages-long sentences without stopping. Please make the manuscript more easily read.
--- Especially in this case, since the Results and the Discussion are separated, I think discussion combining several results from different analytical methods should be included. I am afraid that the present version basically discussing each result separately. (e.g. Is there any trend that can be noticed when three major components of wood are discussed together?)
Response: Thank you for this suggestion. Discussion part is revised by adding sub sessions on the basis of different analysis. The results of certain analysis can be discussed together brough under same sub-session.
Additional minor comments:
- There are many small mistakes in writings, so close proofreading is the must: e.g. typos (which appears especially in the reference list) and mis-writings, mistakes in capitalization and italicization rules, not using superscripts and subscripts where needed, numbering order of the cited references, etc. Please check the sentences in line389 and line 395. Are they expressing what the authors wanted to say correctly? (I am suspicious that some essential words are missing from these sentences.)
Response: The sentences 389-395 have been revised by mentioning monolignol composition data. The typos and grammatical mistakes have been corrected using a licensed version of a professional software. The italics rules for scientific names have been corrected throughout manuscript.
- Regarding the G peak and S peak tagged in Figure 2. Couldn't the descriptions would be more specific, namely the exact compound names?
Response: Figure 2 have been revised as per the suggestion.
- I personally think the result and the discussion presented in the current manuscript is not enough to mention threoβ-O-4 structure may dominate in Leucaena
Response: The discussion part revised by removing ‘threo’. Since wood is enriched with S lignin, β-O-4 aryl ether linkages could be more and therefore, this part has retained.
- Where is the result of lignin content determined by acetyl bromide methods?
Response: The results of this analysis was removed during manuscript finalization. Therefore, acetyl bromide description has been removed from the material and methods.
Reviewer 2 Report
Comments and Suggestions for Authors
The author evaluated the the chemical composition of the secondary xylem of juvenile and adult wood from the branch and main trunk of Leucaena leucocephala, and also got some valuable information. Unfortunately, the author does not correlate these chemical composition to the function and/or specific traits of the wood, thus I suggest the author to link the chemical composition to the function of wood.
other comments:
1. These are many grammar error, the author should corrected them in the whole manuscript. For example, Line 83 the number of the unit (cm-1) should be superscript, which could be found in whole manuscript.
2. In Line 46, the sentence was incomplete, I do not know what the mean of this sentence.
3. I suggest the author put the table 1 as supplementary materials.
4. From the table 2-4, although the author indicated the results of statistical analysis, but I also suggest the author to use letter to indicated them.
Besides, the author can use the figure to instead the table to show the results more clear.
5. the formation of the manuscript was not consistent such as Line 353, Line 362 and Line 373. please correct them.
Comments on the Quality of English LanguageModerate editing is required.
Author Response
- The author evaluated the the chemical composition of the secondary xylem of juvenile and adult wood from the branch and main trunk of Leucaena leucocephala, and also got some valuable information. Unfortunately, the author does not correlate these chemical composition to the function and/or specific traits of the wood, thus I suggest the author to link the chemical composition to the function of wood.
Response: we agree with reviewer’s remarks on importance of chemistry and functional dynamics of wood traits within the tree. We have added a new sub session in the discussion describing the major differences noticed between branch and main trunk wood (two different organs) and their possible reasons with reference to growth dynamics.
other comments:
- These are many grammar error, the author should corrected them in the whole manuscript. For example, Line 83 the number of the unit (cm-1) should be superscript, which could be found in whole manuscript.
Response: The manuscript have been corrected for grammatical mistakes. The unit cm-1 is corrected throughout the manuscript.
- In Line 46, the sentence was incomplete, I do not know what the mean of this sentence.
Response: Thank you very much for finding this important correction. During final formatting, some sentences and its part went missing. The missing sentence and part of the sentence have been added
- I suggest the author put the table 1 as supplementary materials.
Response: As suggested, table 1 have been represented as supplementary material
- From the table 2-4, although the author indicated the results of statistical analysis, but I also suggest the author to use letter to indicated them.
Response: The statistical difference have been indicated in table 4 with asterisk. Letters usually used to represent the levels of significances (confidence levels at 95%, 98%, 99% etc). Here we represented only 95% confidence level ( α=0.05) and therefore putting letters will imply the same message as asterisk.
- Besides, the author can use the figure to instead the table to show the results more clear.
Response: Thank you for your suggestion. To get a better clarity on differences, we prefer to keep the digits in tabular form.
- The formation of the manuscript was not consistent such as Line 353, Line 362 and Line 373. please correct them.
Response: The Sentences have been revised to bring consistency in the statements describing S/G ratio and their degradability properties during pulping process.
Round 2
Reviewer 1 Report
Comments and Suggestions for Authors
Firstly, I would like to appreciate for your sincere responses to my earlier comments.
1 Regarding my previous comment No.3, it might be my fault, but it seems that you are not understanding the point I was trying to make. Regarding Fig. 1, there are four curves. Each upper and lower two seem to be grouped together, but I cannot figure out what is the difference in curves within each WP and KL groups? I would like to ask this point to be revised and clarified.
2 Same as above, you are not understanding my point in previous comment No. 7. Each peak in the chromatograph basically represents a specific compound, and “guaiacyl” and “syringyl” are not compound names. (They are only names of functional groups.) If your understanding of my comment is still not clear, please refer to, for example, Fig. 6.4.3a, b. and Table 6.4.2. in the chapter “Thioacidolysis” in “Methods in Lignin Chemistry” edited by Lin and Dence.
3 There are still many minor errors in the writings. Especially, the word “Klason lignin” is misspelled in the revised version. (There should only be one “s” and not two.)
The followings are several errors, some of those might be difficult to point out by a non-specialist (person not working in this research area).
---(l. 52) Isn’t the reference [6,6] should be [6,7]?
---(l. 392) Please be more specific and accurate about what you are comparing. From the present sentence, it seems to me that you are comparing juvenile and “mature” wood in main trunk with mature wood. If the latter “mature wood” is mentioned as of “branch”, it is acceptable…
---(l. 465, 467) I think “sylylated/sylated” should be “silylated”, but I am not that certain.
---(l. 511) “d” needs to be capitalized, as “CDCl3”.
---(Ref. No. 16) I think it should be “fiber” (US-style. Not British.)
---(Ref. No. 25) It should be Lundquist, K. (Not an “a”.)
---(Ref. No. 49) It should be: Akiyama, T., Matsumoto, Y., Okuyama, T…. (Not an “o”.) Additionally, “o” is missing from the word “threo”.
4 I have forgotten to point this out previously. What kind of wood powder were used to isolate milled wood lignin (Bjorkman lignin)? Please additional description somewhere around l. 496.
Comments on the Quality of English Language
I still think minor editing on capitalization and italicization (erythro and threo officially should be in italic) rules is essential. Also, Journal Editorial need to check the order of numbering the reference list in the text, since the previous Table 1 is now changed to supplemental materials.
Author Response
Thank you very much for the reviewer for critical evaluation and important suggestions to improve the quality of the manuscript. We have revised the manuscript according to the suggestions from the reviewer. Please find the brief details of response to reviewer’s comments.
- Regarding my previous comment No.3, it might be my fault, but it seems that you are not understanding the point I was trying to make. Regarding Fig. 1, there are four curves. Each upper and lower two seem to be grouped together, but I cannot figure out what is the difference in curves within each WP and KL groups? I would like to ask this point to be revised and clarified.
Response: I apologize for not understanding correctly the point indicated by reviewer. The curves belong to branch wood and main trunk wood samples (The upper spectra are from wood powder of branch and main trunk and there are no differences between them). Now I have labelled them to get more clarity.
- Same as above, you are not understanding my point in previous comment No. 7. Each peak in the chromatograph basically represents a specific compound, and “guaiacyl” and “syringyl” are not compound names. (They are only names of functional groups.) If your understanding of my comment is still not clear, please refer to, for example, Fig. 6.4.3a, b. and Table 6.4.2. in the chapter “Thioacidolysis” in “Methods in Lignin Chemistry” edited by Lin and Dence.
Response: Thank you very much for more clarifications on the correction. The figure 2 has been corrected according to the suggestion by reviewer. Peaks have numbered and their chemical structural information have been provided in the legend.
- There are still many minor errors in the writings. Especially, the word “Klason lignin” is misspelled in the revised version. (There should only be one “s” and not two.)
Response: The suggested correction has been made throughout manuscript
- The followings are several errors, some of those might be difficult to point out by a non-specialist (person not working in this research area).
Response: Thank you very much for the keen observations by the reviewer on this important correctios. We have made corrections for all the suggestions made by reviewer.
- ---(l. 52) Isn’t the reference [6,6] should be [6,7]?
Response: The citation number has been corrected.
- 392) Please be more specific and accurate about what you are comparing. From the present sentence, it seems to me that you are comparing juvenile and “mature” wood in main trunk with mature wood. If the latter “mature wood” is mentioned as of “branch”, it is acceptable…
Response: Thank you for bringing this information on the controversial nature of two consecutive sentences. To my understanding reviewer was pointed out that the first sentence says there are differences in chemical composition and latter says chemical composition variation pattern is similar. Considering this controversial nature and the continuity of first and third sentences, the second sentence (The pattern of changes….manner) has been removed.
- (l. 465, 467) I think “sylylated/sylated” should be “silylated”, but I am not that certain.
Response: The suggested correction has been made in the materials and methods
- (l. 511) “d” needs to be capitalized, as “CDCl3”.
Response: The suggested correction has been made in the e materials and methods.
- (Ref. No. 16) I think it should be “fiber” (US-style. Not British.)
Response: The suggested correction has been made in the reference
- (Ref. No. 25) It should be Lundquist, K. (Not an “a”.)
Response: The suggested correction has been made in the reference
- No. 49) It should be: Akiyama, T., Matsumoto, Y., Okuyama, T…. (Not an “o”.) Additionally, “o” is missing from the word “threo”.
Response: The suggested correction has been made in the reference
- I have forgotten to point this out previously. What kind of wood powder were used to isolate milled wood lignin (Bjorkman lignin)? Please additional description somewhere around l. 496.
Response: Milled wood lignin was extracted from the extractive wood powder. This information is provided in the materials and methods.
Reviewer 2 Report
Comments and Suggestions for Authors
All the issues I concerned were solved. I recommend it for publishing.
I also suggest the author the check the whole manuscript to avoid grammar error.
Author Response
Thank you very much for the evaluation, for providing important suggestions and comments to improve the quality of our manuscript, and for accepting the revised version.
Round 3
Reviewer 1 Report
Comments and Suggestions for Authors
1 Regarding the peak description in Fig. 2, it is now much better. However, I couldn’t understand what the authors want to say with “50/50”. Was this erythro/threo ratio experimentally determined? Or is it theoretical? In the first place, is it necessary?
2 The sentence in line 392 has not been revised as I had asked for. The subjects of comparison are strange (or ambiguous), and this sentence cannot be interpreted.
3 The authors have corrected “fibre” spelling those were unnecessary. Regarding the titles of Refs. No.19 and 38, they seem to be British-style (fibres).
4 Regarding the response for Comment No.4, I think the revision is not enough. Since this manuscript deal with the difference between branch/trunk, and juvenile/adult, the description of the origin of MWL as “wood powder” is not enough. I do know that large amounts of wood powder are necessary to obtain MWL, so if you have used wood powder from a whole tree (not specific parts) even is no problem, but please do mention so clearly.
Comments on the Quality of English LanguageThe followings are those of the repeat of my previous comment:
I still think minor editing on capitalization and italicization rules is essential. (Also, Journal Editorial need to check the order of numbering the reference list in the text, since the previous Table 1 is now changed to supplemental materials.)
Author Response
Thank you very much for the critical suggestions to improve the manuscripr version 2. We have revised the manuscript according to the suggestions from the reviewer. The list of changes made in the version 3 is listed below
- Regarding the peak description in Fig. 2, it is now much better. However, I couldn’t understand what the authors want to say with “50/50”. Was this erythro/threo ratio experimentally determined? Or is it theoretical? In the first place, is it necessary?
Response: The erythro/threo 50/50 was theoretical. It is removed in the revised version.
- The sentence in line 392 has not been revised as I had asked for. The subjects of comparison are strange (or ambiguous), and this sentence cannot be interpreted.
Response: The sentence have been corrected by mentioning branch wood in the end of the sentence. Now the sentence is ‘Both the juvenile and mature wood in main trunk xylem showed higher content of holocellulose and α-cellulose content, less lignin and low S/G ratio compared to that of branch wood.
- The authors have corrected “fibre” spelling those were unnecessary. Regarding the titles of Refs. No.19 and 38, they seem to be British-style (fibres).
Response: The suggested references where spelling of fibre has been corrected as per the suggestion.
- Regarding the response for Comment No.4, I think the revision is not enough. Since this manuscript deal with the difference between branch/trunk, and juvenile/adult, the description of the origin of MWL as “wood powder” is not enough. I do know that large amounts of wood powder are necessary to obtain MWL, so if you have used wood powder from a whole tree (not specific parts) even is no problem, but please do mention so clearly.
Response: More details of the samples (juvenile/Mature from branch and main trunk) and amounts of wood powder for each sample used for extraction have been added in the methodology part.
- The followings are those of the repeat of my previous comment:
I still think minor editing on capitalization and italicization rules is essential. (Also, Journal Editorial need to check the order of numbering the reference list in the text, since the previous Table 1 is now changed to supplemental materials.)
Response: Capitalization for main title (first letter of each main word) and italicization of subheadings have been made in the revised version. Table numbers were corrected in the main text after keeping table 1 in supplementary material.